# Azithromycin Adsorption onto Different Soils

Raquel Cela-Dablanca [1,*], Ana Barreiro [1], Lucía Rodríguez-López [2], Paula Pérez-Rodríguez [2], Manuel Arias-Estévez [2], María J. Fernández-Sanjurjo [1], Esperanza Álvarez-Rodríguez [1] and Avelino Núñez-Delgado [1]

[1]  Department Soil Science and Agricultural Chemistry, Engineering Polytechnic School, Universidad Santiago de Compostela, 27002 Lugo, Spain
[2]  Soil Science and Agricultural Chemistry, Faculty Sciences, University of Vigo, 32004 Ourense, Spain
*   Correspondence: raquel.dablanca@usc.es

**Abstract:** The antibiotic azithromycin (AZM) is one of the most persistent in the environment, with potential to cause serious health and environmental problems. As some polluting discharges containing this antibiotic can reach the soil, it is clearly relevant determining the ability of soils with different characteristics to retain it. In this research, AZM adsorption and desorption were studied for a variety of soils, using batch-type experiments. The results show that, at low doses of antibiotic added (less than or equal to 50 $\mu$mol L$^{-1}$), the adsorption always reached 100%, while when higher concentrations were added (between 200 and 600 $\mu$mol L$^{-1}$) the highest adsorption corresponded to soils with higher pH values. Adsorption data were fitted to the Linear, Langmuir and Freundlich models, with the latter showing the best fit, in view of the determination coefficient. No desorption was detected, indicating that AZM is strongly adsorbed to the soils evaluated, suggesting that the risks of environmental problems due to this contaminant are minimized for these edaphic media. These results can be considered relevant with respect to risk assessment and possible programming of measures aimed at controlling environmental contamination by emerging contaminants, especially from the group of antibiotics, and in particular from AZM.

**Keywords:** antibiotic; emerging contaminants; adsorption; desorption; environment





## 1. Introduction

Azithromycin (AZM) is a semi-synthetic broad spectrum antibiotic belonging to the subclass of second-generation macrolides [1]. It is used to treat bacterial infections in infants and in people with weaker immune systems, among other diseases [2]. In the US, AZM is among the first-line agents prescribed for infectious diseases [3]. In 30 European countries, this group of antimicrobials together with beta-lactams, lincosamides, streptogramins and tetracyclines accounted for 83.5% of total antibiotic sales in 2013 [4]. This antibiotic is on the DU75 list (among the 75% of the most consumed antibiotics) in 24 of 46 countries in the European area [5]. In addition, during 2020 the use of AZM increased significantly in Spain [6].

When AZM, as well as other antibiotics and other emerging contaminants, reach the environment through polluting discharges, it is considered a cause of concern, especially taking into account the high concentrations detected in aquatic environments [7–10]. In fact, there are several routes for antibiotics to reach the environment as pollutants, but the main one is through wastewater [11]. The cause is that, after their administration as drugs, these antimicrobials are partially metabolized and released through urine and feces [12]. Specifically regarding AZM, 75% of it is excreted after being administered [13], reaching wastewater treatment plants.

The efficacy of these treatment plants is dependent on factors such as the type of treatment, or the nature and properties of the antibiotic to be treated. Specifically, AZM belongs to a group of antibiotics of special relevance in view of its prevalence in the

environment, due to its persistence and resistance to biological degradation [14]. The effectiveness of different treatments applied in wastewater treatment plants to retain this antibiotic was studied by Mirzaei et al. [15], finding that the efficiency of one of the treatments ranged from 0% to 74.9%, while another treatment was not effective in removing AZM. In addition, in a study carried out in twelve wastewater plants in China [16], it was found that AZM was one of the antibiotics that appeared most frequently, and it was affected by the lowest elimination rate efficiency, specifically 6.3%.

Antibiotics that cannot be eliminated after wastewater treatment persist in these waters and/or end up in the sludge generated by this treatment, reaching agricultural soils through the application of irrigation wastewaters and/or biosolids [17]. In a study by Rodríguez-Mozaz et al. [18], the authors found the presence of AZM in a range of 45.2–597.5 ng L$^{-1}$, with Portugal showing the highest concentrations of this antibiotic, while the minimum values were obtained in Cyprus. In North America, Europe and elsewhere, the use of biosolids as agricultural soil amendment is permitted [19], and specifically, in the EU-27, 53% of the total sludge produced is recycled in agriculture directly or after composting [20]. In some countries such as Denmark, France, the Walloon region of Belgium, Ireland, Spain and the United Kingdom, more than half of the sludge production ends up on agricultural land, while in other countries such as Finland, the Netherlands, Slovakia, Greece and Slovenia, the amounts are less than 5% [21].

The persistence of antibiotics in these sludges can lead to contamination of the soil and other environmental compartments, such as surface water and groundwater by leaching or runoff processes, or even enter the food chain through vegetables grown in contaminated soils [22]. Walters et al. [23] carried out a study on the persistence of antibiotics in mixtures of biosolids with soils and found that AZM had a half-life of 408–3466 days. Of special concern as regards health risks related to the presence of antibiotics in soil are the emergence and spread of antibiotic resistance in pathogenic bacteria [24].

The characterization of the retention and release processes of these antibiotics in soils amended with biosolids is essential to evaluate their transport and the risks associated with their presence [25]. The behavior of these contaminants in the soil depends on edaphic characteristics, such as organic carbon and clay contents, texture and pH [26] and on antibiotics properties, such as hydrophobicity, solubility and molecular structure [27] and their degrees of ionization [28]. The dissociation constant (pK$_a$) is a parameter to predict the ionization state of a molecule with respect to pH [29]. The antibiotic AZM has two pk$_a$ values, pk$_{a1}$ = 8.74 and pk$_{a2}$ = 9.45, causing that in most soils the molecule behaves as cation [30], however, some authors indicate only one pk$_a$ value, being 7.25 [31] or 8.96 [32]. This antibiotic presents multi-basic amines, having pk$_a$ values that could allow suffering protonation in a rather specific (physiological) pH range [33].

It is also interesting to determine and bear in mind the sensitivity of these molecules to different environmental conditions that could affect their degradation [34].

Taking all of the above into account, the objective of this work is to determine AZM adsorption and desorption on/from soils with different physical-chemical properties. This will be key to evaluating the risk that the presence of this antibiotic may pose in the soil environment, as well as, due to its eventual mobility, the risk of pollution affecting to other different environmental compartments.

## 2. Materials and Methods

### 2.1. Soils

A total of 21 soils were selected for the study, all of them previously sampled at different areas of Galicia (NW Spain). Six of the soils correspond to plots planted with corn/maize (designated with codes from M1 to M6), while twelve soils were from vineyards (from VO1 to VO5 -Ourense province-, and from VP1 to VP7 -Pontevedra province), and three were forest soils, of which one was pine forest (FP), another eucalyptus (FE) and another oak (*Q. robur*) (FR). These soils were selected based on their pH values and organic

matter contents. Table S1 (Supplementary Material) shows some characteristics of the areas where the soil samples were collected.

Each soil sample was made up of 10 subsamples collected in a zig-zag pattern in the surface layer (0–20 cm). Once collected, the samples were dried at 40 °C to constant weight, and then sieved through a 2 mm diameter sieve and stored until analysis. The soils used in this work were previously characterized by Cela-Dablanca et al. [35]. Table S2 (Supplementary Material) shows the main physical-chemical properties of the selected soils.

### 2.2. Chemical Reagents

The AZM used was supplied by Sigma-Aldrich (Barcelona, Spain). Figure S1 (Supplementary Material) shows the molecular structure of AZM. Potassium phosphate (purity $\geq$ 99.5%) and acetonitrile (purity $\geq$ 99.9%) used for HPLC were supplied by Fisher Scientific (Madrid, Spain) and $CaCl_2$ (95% purity) by Panreac (Barcelona, Spain). To carry out HPLC determinations, all solutions were prepared with milliQ water (Millipore, Madrid, Spain).

### 2.3. Sorption and Desorption Experiments

Batch-type experiments were carried out to study the adsorption/desorption of AZM on/from the different soils. For this, 2 g of each sample were weighed, then adding 5 mL of a solution with different concentrations of the antibiotic (2.5, 5, 10, 20, 30, 40, 50, 200, 400, and 600 $\mu$mol L$^{-1}$), also containing 0.005 M $CaCl_2$ as background electrolyte. The suspensions were shaken in the dark for 48 h (time enough to reach equilibrium, according to previous kinetic tests) using a rotary shaker. These suspensions were then centrifuged at 4000 rpm for 15 min (G force: 1931.91). The resulting supernatants were filtered through 0.45 $\mu$m nylon syringe filters. Finally, the antibiotic concentrations in the equilibrium solution were determined by HPLC-UV with a LPG 3400 SD equipment (Thermo-Fisher, Waltham, MA, USA). The quantification method, as well as further details regarding adsorption and desorption experiments, are shown in Supplementary Material.

### 2.4. Data Treatment

The experimental data obtained in the adsorption/desorption tests were adjusted to the Freundlich (Equation (1)), Langmuir (Equation (2)) and Linear (Equation (3)) models [36].

$$q_e = K_F C_{eq}^n \tag{1}$$

$$q_e = \frac{K_L \, C_{eq} q_m}{1 + K_L C_{eq}} \tag{2}$$

$$q_e = K_d C_{eq} \tag{3}$$

where $q_e$ (expressed in $\mu$mol kg$^{-1}$) is the amount of antibiotic retained onto the soil (calculated as the difference between the concentration added and that remaining in the equilibrium solution); $K_F$ (Ln $\mu$mol$^{1-n}$ kg$^{-1}$) is the Freundlich constant related to the adsorption capacity; $C_{eq}$ ($\mu$mol L$^{-1}$) is the concentration of antibiotic present in the solution at equilibrium; $n$ (dimensionless) is a parameter of the Freundlich model associated with the degree of heterogeneity of the adsorption; $K_L$ (L $\mu$mol$^{-1}$) is the Langmuir adsorption constant; $q_m$ ($\mu$mol kg$^{-1}$) is the maximum adsorption capacity according to the Langmuir model; and $K_d$ (L kg$^{-1}$) is the partition coefficient in the linear model.

In addition, soil properties were correlated with the parameters obtained in the adjustments to the adsorption models, determining the Pearson correlation coefficients.

The SPSS Statistics 21 software was used to carry out the adjustment of the data derived from the adsorption experiments to the Langmuir, Freundlich and Linear models, as well as any further statistical analysis.

## 3. Results and Discussion

### 3.1. Adsorption

Figure 1 shows relations among equilibrium concentrations and AZM adsorption for the different soils. The results indicate that the corn/maize soils have a maximum adsorption value of 1256.77 µmol kg$^{-1}$, similar to that found in vineyard soils, which specifically reach 1229.38 and 1318.21 µmol kg$^{-1}$ in granite soils and in slate/schist soils, respectively. Regarding forest soils, AZM adsorption is much lower than in crop soils, with maximum value (228.84 µmol kg$^{-1}$) found in the eucalyptus soil sample. Regarding the minimum adsorption scores, for the highest AZM concentration added they were the following: 754.04 µmol kg$^{-1}$ in soils with corn cultivation, 690.96 µmol kg$^{-1}$ in granite vineyard soils, 401.27 µmol kg$^{-1}$ in slate/schist vineyard soils, and 187.29 µmol kg$^{-1}$ in forest soils.

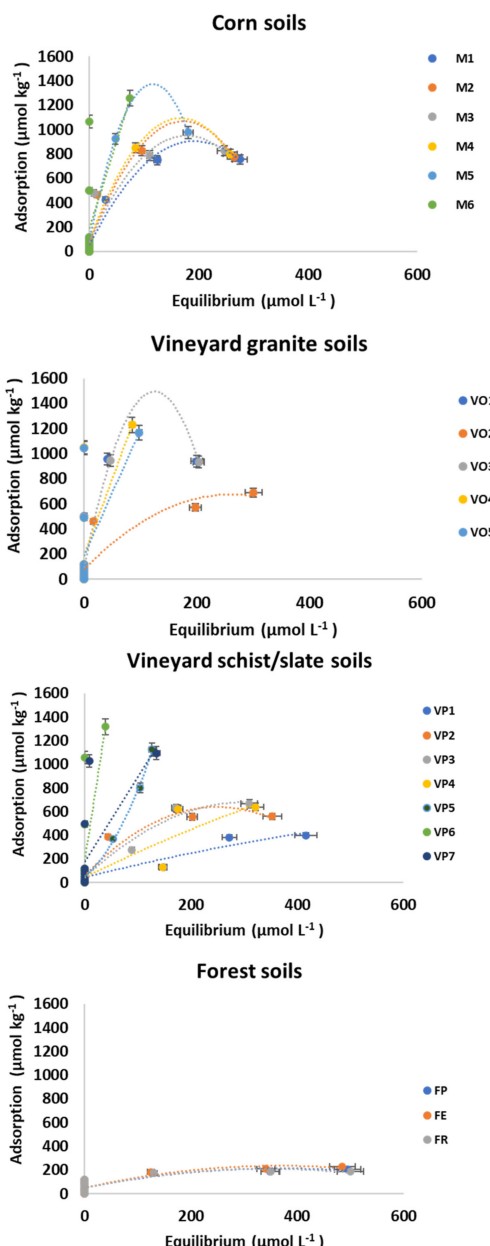

**Figure 1.** AZM adsorption curves for the different soils studied. M: corn/maize soils, VO: vineyard soils (Ourense Province), VP: vineyard soils (Pontevedra province), FP: forest soil (*P. pinaster*), FE: forest soil (*E. nitens*), FR: forest soil (*Q. robur*), 1–7: different soil samples. Error bars indicate standard deviation.

Figure 2 shows that the percentage adsorption values were 100% for all the soils when the antibiotic concentrations added were in the range of 2.5–50 µmol L$^{-1}$, while marked differences are observed among the different soils when the highest AZM concentrations (200–600 µmol L$^{-1}$) were added. To note that, in general, in this last range of concentrations added the percentage of adsorption decreases as the concentration of antibiotic added increases, probably due to the saturation of the adsorption sites [37]. Considering specific values, most maize soils show high adsorption percentages, ranging between 52.29% and 100%, very close to those obtained in granite vineyard soils, which range between 47.92% and 100%. Adsorption on slate/schists vineyard soils varied over a broader range (from 26.12% to 100%). On the other hand, forest soils were those that presented the lowest adsorption scores (between 13.25% and 37.27%).

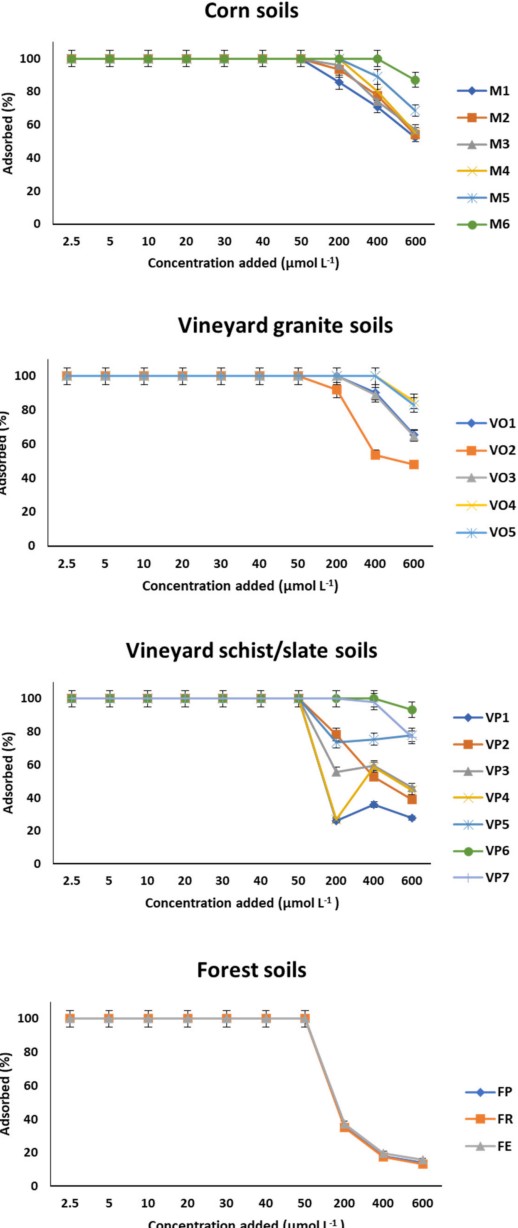

**Figure 2.** AZM adsorption percentages for the soils studied, as a function of the AZM concentration added. M: corn/maize soils, VO: vineyard soils (Ourense Province), VP: vineyard soils (Pontevedra province), FP: forest soil (*P. pinaster*), FE: forest soil (*E. nitens*), FR: forest soil (*Q. robur*), 1–7: different soil samples. Error bars indicate standard deviation.

When comparing adsorption data with soil characteristics (Table S2, Supplementary Material) it is clear that the soils showing higher adsorption were those having higher pH values (M5, M6, VP6 and VP7). In fact, adsorption data were correlated with soil properties, indicating that AZM adsorption correlates positively and significantly with soil pH (r = 0.562 and $p < 0.01$) (Table 1). To note that pH is one of the parameters having greater influence on antibiotic-adsorbent interactions, since it simultaneously affects the chemical speciation of the pollutants and adsorbent surfaces [38]. Both antibiotics and soil components have functional groups that may suffer protonation/deprotonation, depending on the pH of the solution. This makes it possible for there to have positive, negative or neutral charges on the reactive surfaces, and therefore allows the formation of different types of bonds [39]. The $pK_a$ of AZM is approximately between 8.6–9.5, so in an acid medium this antibiotic is protonated [40]. In the pH range of the soils included in this study (4.68–8.02), AZM has positively charged functional groups, thus favoring electrostatic interactions with the negatively charged surfaces of minerals and organic matter [41,42].

**Table 1.** Correlations between AZM adsorption values and soil properties. OM: organic matter; $Al_{ox}$: Non-crystalline aluminum (extracted with ammonium oxalate).

| Soil Property | Correlation Coefficient (r) | Significance Level (*p*) |
|:---:|:---:|:---:|
| pH | 0.562 | 0.01 |
| OM | 0.530 | 0.05 |
| $Al_{ox}$ | −0.43 | 0.05 |

Soil organic matter and the non-crystalline Fe and Al components have variable charge, with negative charge increasing as the pH rises, favoring higher AZM adsorption to these charged surfaces, as AZM is mainly present as a divalent cation. For this reason, forest soils, despite the fact that they have a high content of both organic matter and non-crystalline minerals, present low AZM adsorption when the antibiotic is added at concentrations above 200 μmol $L^{-1}$. This is due to the fact that their pH is very low and there are few negative charges. However, soils having lower organic matter and non-crystalline minerals contents (M5, M6, VO4, VP6, VP7), show higher adsorption than forest soils, because they have a pH >5.8 and higher presence of negative charges. This would explain the significant ($p < 0.05$) and negative correlation obtained between the maximum adsorption of each soil with the organic matter content (r = −0.53) and the total non-crystalline Al extracted with ammonium oxalate (r = −0.43) (Table 1). Other authors also found no positive correlations between non-crystalline Fe components and AZM adsorption [43]. However, in our study we not found any correlations (positive or negative) with this parameter. The high AZM adsorption obtained in the current study for various agricultural soils (Figures 1and 2) is consistent with that found in other researches carried out with high concentrations of this antibiotic in amended crop soils, which on the other hand showed that this drug does not cause toxicity in crops or soil microorganisms [44,45].

The adsorption data were fitted to the Linear, Freundlich and Langmuir models, which are the most commonly used to establish equilibrium relations between an adsorbent and an adsorbate, or between the amount adsorbed to a solid phase and that which remains in solution at a given temperature under equilibrium conditions [30].

Table 2 shows the adsorption parameters obtained from the fit of the experimental data to the Freundlich, Langmuir and Linear adsorption models.

Taking into account the values of the coefficient of determination ($R^2$), it can be considered that the model showing a better fit was the Freundlich equation, with $R^2$ greater than 0.85 for the 33% soils, while the Langmuir model obtained a $R^2$ value > 0.85 for 23% of the soils, and for the Linear model this porcentage decrease until 9%. However, in the Freundlich and in the Langmuir model, very high error values were associated with the estimation of the parameters in many cases, so they would not satisfactorily explain AZM adsorption in these soils. To note that some authors indicate that AZM follows linear

adsorption models in soils amended with biosolids [46], although the soils in the current study have higher pH values.

**Table 2.** Fitting of the adsorption data to the Freundlich, Langmuir and Linear models. $K_F$ expressed in $L^n \mu mol^{1-n} kg^{-1}$; $n$ is dimensionless; $K_L$ expressed in $L \mu mol^{-1}$; $q_m$ expressed in $\mu mol kg^{-1}$; $K_d$ expressed in $L kg^{-1}$; -: error values too high for fitting.

| | Freundlich | | | | | Langmuir | | | | | Linear | | |
|---|---|---|---|---|---|---|---|---|---|---|---|---|---|
| Soil | $K_F$ | Error | $n$ | Error | $R^2$ | $K_L$ | Error | $q_m$ | Error | $R^2$ | $K_d$ | Error | $R^2$ |
| M1 | 95.986 | 50.522 | 0.387 | 0.103 | 0.89 | 0.028 | 0.012 | 901.315 | 100.849 | 0.94 | 3.403 | 0.55 | 0.67 |
| M2 | 341.346 | 94.081 | 0.161 | 0.057 | 0.95 | 0.097 | 0.037 | 852.124 | 60.454 | 0.96 | 3.653 | 0.77 | 0.52 |
| M3 | 354.449 | 72.119 | 0.159 | 0.042 | 0.97 | 0.167 | 0.058 | 837.409 | 49.227 | 0.97 | 4.026 | 0.72 | 0.62 |
| M4 | - | - | - | - | - | - | - | - | - | - | 3.787 | 0.9 | 0.44 |
| M5 | - | - | - | - | - | - | - | 995.768 | 258.357 | 0.8 | 6.309 | 1.42 | 0.49 |
| M6 | 9.471 | 0 | 1.136 | 0.073 | 0.31 | - | - | 1600 | 0 | 0.31 | 16.97 | 5.08 | 0.31 |
| VO1 | - | - | - | - | - | - | - | - | - | - | 5.456 | 1.43 | 0.39 |
| VO2 | 314.979 | 91.637 | 0.128 | 0.057 | 0.94 | 0.143 | 0.085 | - | - | 0.94 | 2.532 | 0.41 | 0.66 |
| VO3 | - | - | - | - | - | - | - | - | - | - | 5.376 | 1.35 | 0.42 |
| VO4 | 9.219 | 0 | 1.101 | 0.072 | 0.31 | - | - | 1600 | 0 | 0.31 | 14.46 | 4.36 | 0.31 |
| VO5 | 8.87 | 0 | 1.066 | 0.073 | 0.27 | - | - | 1554.29 | 0 | 0.27 | 11.99 | 3.78 | 0.27 |
| VP1 | - | - | 0.829 | 0.365 | 0.79 | - | - | - | - | - | 1.08 | 0.13 | 0.79 |
| VP2 | 203.771 | 95.973 | 0.178 | 0.089 | 0.93 | 0.041 | 0.026 | 608.522 | 66.977 | 0.93 | 1.952 | 0.32 | 0.65 |
| VP3 | - | - | 0.552 | 0.182 | 0.91 | - | - | - | - | - | 2.546 | 0.26 | 0.86 |
| VP4 | - | - | 0.872 | 0.418 | 0.77 | - | - | - | - | - | 2.149 | 0.29 | 0.77 |
| VP5 | - | - | 1.326 | 0.213 | 0.98 | - | - | - | - | - | 8.274 | 0.4 | 0.97 |
| VP6 | 10.24 | 0 | 1.327 | 0.082 | 0.36 | - | - | 1600 | 0 | 0.36 | 33.87 | 9.59 | 0.36 |
| VP7 | - | - | - | - | – | - | - | 1099.382 | 190.379 | 0.84 | 8.581 | 2.56 | 0.3 |
| FP | - | - | - | - | – | - | - | 209.883 | 80.292 | 0.44 | 0.493 | 0.11 | 0.2 |
| FR | - | - | - | - | – | - | - | 192.928 | 73.297 | 0.4 | 0.466 | 0.11 | 0.1 |
| FE | - | - | - | - | – | - | - | 242.635 | 87.674 | 0.54 | 0.559 | 0.11 | 0.33 |

The values of the distribution coefficient of the linear model ($K_d$), a parameter related to the adsorption intensity, range between 3.403 and 16.973 $L kg^{-1}$ in maize soils, between 2.532 and 14.463 $L kg^{-1}$ in vineyard granite soils, between 1.080 and 33.867 $L kg^{-1}$ in slate/schists vineyards, and between 0.466 and 0.599 $L kg^{-1}$ in forest soils. The lower values of forest soils indicate greater AZM mobility in them [47]. These values are higher than those obtained in previous studies for sulfonamides [48], but lower than those obtained for tetracyclines [49], indicating that interactions with these soils that give rise to AZM adsorption are stronger than those of sulfonamides and weaker than those of tetracyclines. As for the values of the Freundlich affinity coefficient ($K_F$), related to the soil adsorption capacity, they indicate that corn and vineyard soils on granite are the ones with the highest affinity for AZM adsorption.

Bearing in mind that the lower the value of $n$, the more heterogeneous the adsorption surface [50], and also that values of this parameter being between 1 and 10 indicate favorable conditions for adsorption [51], the fact that some of the maize and vineyard soils in this study are the ones with the lowest $n$ values would indicate that they are the ones with a more heterogeneous surface. As comparison, Bao et al. [52] obtained lower $n$ values for tetracycline in forest than in agricultural soils. Regarding the Langmuir model, the lowest values of the $q_m$ parameter (the maximum Langmuir's adsorption capacity) corresponded to some of the forest soils, which would confirm that these soils are the ones with the lowest adsorption capacity.

*3.2. Desorption*

The desorbed AZM concentrations were lower than the detection limit in all the soils studied and for all the antibiotic concentrations added. These results indicate that AZM is adsorbed very strongly onto these soils, causing that the retention process could be

considered almost not reversible under the conditions of this study. These results are consistent with previous research indicating strong AZM adsorption in both biosolids [42] and biosolids-amended soils [30].

## 4. Conclusions

When AZM concentrations of up to 50 μmol $L^{-1}$ are added to the soils used in this research, its adsorption was 100% in all of them, while when the concentration added was equal to or greater than 200 μmol $L^{-1}$, some differences were observed, with those soils having higher pH showing higher adsorption. Adsorption data fitted better to the Freundlich model, in the sense of presenting higher determination coefficients. Regarding AZM desorption, its concentrations were always lower than the detection limit, indicating that the antibiotic was adsorbed in a very strong manner to the soils studied. These results can be considered relevant in terms of assessment of risks of pollution due to AZM, both in soils and in other environmental compartments to which the antibiotic could migrate. It should be noted that, although AZM is present in the environment at lower concentrations than those found in this research, its consumption has clearly increased in recent years, which makes the current work relevant as regard increasing the knowledge on the factors that influence this fact, and facilitating the future development of solutions to the problem. In addition, taking into account that desorption is practically absent from the soils studied, it is suggested that these edaphic environments could help to prevent AZM leaching/transportation to other environmental compartments such as water bodies, especially in case that the adsorption capacity of these soils could be increased by means of low-cost sorbents.

**Supplementary Materials:** The following supporting information can be downloaded at: https://www.mdpi.com/article/10.3390/pr10122565/s1. Table S1: Basic details corresponding to the different soils used in this work. M: maize (corn) soils; VO: vineyard soils (Ourense province); VP: vineyard soils (Pontevedra province); F: forest soils; Table S2: Values corresponding to the basic parameters determined in the various soils studied. M: maize (corn) soils; VO: vineyard soils (Ourense province); VP: vineyard soils (Pontevedra province); F: forest soils. OC: organic carbon; OM: organic matter; N: nitrogen; eCEC: effective cation exchange capacity; Alox and Feox: Al and Fe extracted with ammonium oxalate; Alpir and Fepir: Al and Fe extracted with sodium pyrophosphate. Average values (*n* = 3), with coefficients of variation always <5%; Table S3: AZM adsorption expressed in μmol $kg^{-1}$ (and in percentage between brackets), for the soils studied, as a function of the concentration of antibiotic added. M: maize (corn) soils; VO: vineyard soils (Ourense province); VP: vineyard soils (Pontevedra province); F: forest soils. Average values (*n* = 3), with coefficients of variation always <5%; Figure S1: Molecular structure of AZM; Figure S2. Molecular structure of AZM with amine groups selected; Figure S3: Selected chromatograms corresponding to AZM adsorption onto soils.

**Author Contributions:** Conceptualization, E.Á.-R., M.J.F.-S., A.N.-D. and M.A.-E.; methodology, E.Á.-R., M.J.F.-S., A.N.-D., M.A.-E., R.C.-D. and A.B.; software, E.Á.-R., L.R.-L., P.P.-R. and R.C.-D.; validation, E.Á.-R., M.J.F.-S., A.N.-D., M.A.-E. and A.B.; formal analysis, R.C.-D., L.R.-L. and P.P.-R.; investigation, E.Á.-R., M.J.F.-S., R.C.-D., L.R.-L., P.P.-R. and A.B.; resources, E.Á.-R., M.J.F.-S. and M.A.-E.; data curation, E.Á.-R., M.J.F.-S., A.N.-D. and A.B.; writing—original draft preparation, E.Á.-R., M.J.F.-S., A.B. and R.C.-D.; writing—review and editing, A.N.-D.; visualization, E.Á.-R., M.J.F.-S., A.N.-D., M.A.-E., R.C.-D., L.R.-L., P.P.-R. and A.B.; supervision, E.Á.-R., M.J.F.-S. and A.B.; project administration, E.Á.-R., M.J.F.-S. and M.A.-E.; funding acquisition, E.Á.-R., M.J.F.-S. and M.A.-E. All authors have read and agreed to the published version of the manuscript.

**Funding:** This research was funded by Spanish Ministry of Science, Innovation and Universities, grant numbers RTI2018-099574-B-C21 and RTI2018-099574-B-C22.

**Institutional Review Board Statement:** Not applicable.

**Informed Consent Statement:** Not applicable.

**Data Availability Statement:** Not applicable.

**Conflicts of Interest:** The authors declare no conflict of interest. The funders had no role in the design of the study; in the collection, analyses, or interpretation of data; in the writing of the manuscript, or in the decision to publish the results.

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
