# Peer review of "Azithromycin Adsorption onto Different Soils"

_processes, doi:10.3390/pr10122565_

Round 1
Reviewer 1 Report
The paper investigates the (de)sorption of azithromycin (AZM) onto different soils using different initial AZM concentrations. Results were fitted to Freundlich, Langmuir and linear models and discussed in context of the respective soil chemical parameters of each soil. The topic is of environmental relevance in order to improve the understanding on the fate of pharmaceuticals in soils. The manuscript is well prepared and easy to read.
I only have a few minor comments and suggest acceptance of the manuscript once they have been addressed:
Lines 78-80: it would be helpful to present an additional molecular structure of the substance, depicting the location of the positive charge. Which groups dissociate?
Line 118: “at 4000 rpm for 15 minutes” please provide how many g this corresponds to (this will depend on rotor settings)
Caption of figure 1: What do the numbers M1-M6 stand for? Please add this information in the figure caption.
All figures: please indicate the meaning of the error bars (standard deviation?) and state it in the caption.
Line 209: “… AZM is mainly present as a divalent cation.” Please indicate this in the molecular structure in the supporting information.
Line 218: “… non-crystalline Al…” And what about Fe?
Table 2 and lines 239-241: R² of Freundlich are lower or equal to R² of Langmuir. Hence, I do not understand this sentence. Please check and clarify.
Conclusions: What is stated here is more or less a summary of the results. However, a conclusion should be more than that. Please state some more environmental implications which can be derived from the results.
Author Response
Comments and Suggestions for Authors
The paper investigates the (de)sorption of azithromycin (AZM) onto different soils using different initial AZM concentrations. Results were fitted to Freundlich, Langmuir and linear models and discussed in context of the respective soil chemical parameters of each soil. The topic is of environmental relevance in order to improve the understanding on the fate of pharmaceuticals in soils. The manuscript is well prepared and easy to read.
RESPONSE: Thank you for your positive comments.
I only have a few minor comments and suggest acceptance of the manuscript once they have been addressed:
Lines 78-80: it would be helpful to present an additional molecular structure of the substance, depicting the location of the positive charge. Which groups dissociate?
RESPONSE: Thank you for your comment. We have added the required information into the manuscript (red fonts).
Line 118: “at 4000 rpm for 15 minutes” please provide how many g this corresponds to (this will depend on rotor settings)
RESPONSE: Thank you for you comment. We added the requested information into the manuscript (red fonts).
Caption of figure 1: What do the numbers M1-M6 stand for? Please add this information in the figure caption.
RESPONSE: Thank you for your comment. We added the information in the manuscript (red fonts).
All figures: please indicate the meaning of the error bars (standard deviation?) and state it in the caption.
RESPONSE: Thank you for your comment. We added the information in the manuscript (red fonts).
Line 209: “… AZM is mainly present as a divalent cation.” Please indicate this in the molecular structure in the supporting information.
RESPONSE: Thank you for your comment. We added the requested information in the Supplementary material (red fonts).
Line 218: “… non-crystalline Al…” And what about Fe?
RESPONSE: Thank you for your comment. We have added de requested information into the manuscript (red fonts).
Table 2 and lines 239-241: R² of Freundlich are lower or equal to R² of Langmuir. Hence, I do not understand this sentence. Please check and clarify.
RESPONSE: Thank you for your comment. We change the information into the manuscript (red fonts).
Conclusions: What is stated here is more or less a summary of the results. However, a conclusion should be more than that. Please state some more environmental implications which can be derived from the results.
RESPONSE: Thank you for your comment. We have added the information into the manuscript (red fonts).
Reviewer 2 Report
1. A good paper covering a wide range of soils with variable properties.
2. Table 2: there are no Freundlich data for forest soils!
3. it appears that the 48hr equilibrium time used is rather extensive. If the AZM parent compound is stable and not degrading under the various challenging conditions (pH of soils), measurements are likely reliable. What happens when the AZM degrades to the metabolites in the 48 hr? Do we have measurements of the metabolites in the supernatant?
4. Are the high AZM concentrations used in this experiment reasonably close to the concentrations found in wastewater treatment plants and residue soils? If so, then these high concentrations would be a good comparison of real world situations.
5. Since AZM is adsorbed strongly to agricultural soils it appears to be encouraging to reduce leaching to deeper profiles, and groundwater. However, there are serious implications to ecological impacts if the AZM doesnt degrade readily in the surface soils such as remaining as a long term source of antibiotic contamination and other organisms developing antibiotic resistance.
Author Response
Comments and Suggestions for Authors
- A good paper covering a wide range of soils with variable properties.
RESPONSE: Thank you for the positive comment.
- Table 2: there are no Freundlich data for forest soils!
RESPONSE: Thank you for your comment. The errors obtained in the adjustment was very high, for these reason we decided to delete the values.
- it appears that the 48hr equilibrium time used is rather extensive. If the AZM parent compound is stable and not degrading under the various challenging conditions (pH of soils), measurements are likely reliable. What happens when the AZM degrades to the metabolites in the 48 hr? Do we have measurements of the metabolites in the supernatant?
RESPONSE: Thank you for your comment. Please, note that Vermillion-Maier et al. (2018) conducted a study of the degradation of AZM, finding that in a time period of 150 days it degraded less than 1%. Sidhu et al. (2019) indicate that, regarding AZM speciation and long-term retention-release, soil pH may have an influence, but that this effect is favored at pH>8, while in our soils the pH values are lower. In addition, Topp et al. (2016) also detected azithromycin metabolites.
- Are the high AZM concentrations used in this experiment reasonably close to the concentrations found in wastewater treatment plants and residue soils? If so, then these high concentrations would be a good comparison of real world situations.
RESPONSE: Thank you for your comment. We may comment that the AZM levels found in different studies are lower than the concentrations we added to carry out the adsorption test, but the adsorption of this antibiotic at those concentrations was 100%, so we decided to increase it to better understand the capacity of these soils to adsorb the antibiotic and be able to make comparisons.
- Since AZM is adsorbed strongly to agricultural soils it appears to be encouraging to reduce leaching to deeper profiles, and groundwater. However, there are serious implications to ecological impacts if the AZM doesnt degrade readily in the surface soils such as remaining as a long term source of antibiotic contamination and other organisms developing antibiotic resistance.
RESPONSE: Thank you for your comment. In this regard, we have to indicate that in a study conducted by Konopka et al. (2022) these authors found that, in terrestrial ecosystems, there is minimal toxicity to both microbial communities and plants used for providing food, and zero toxicity to earthworms. However, over a period of several years and with repeated applications of biosolids, the abundance of antibiotic resistance genes (ARGs) in the soil could increase.